# Double Valorization for a Discard—α-Chitin and Calcium Lactate Production from the Crab *Polybius henslowii* Using a Deep Eutectic Solvent Approach

**DOI:** 10.3390/md20110717

**Published:** 2022-11-16

**Authors:** Colin McReynolds, Amandine Adrien, Arnaud Petitpas, Laurent Rubatat, Susana C. M. Fernandes

**Affiliations:** 1Universite de Pau et des Pays de l’Adour, E2S UPPA, IPREM, CNRS, 64 600 Anglet, France; 2MANTA—Marine Materials Research Group, Universite de Pau et des Pays de l’Adour, E2S UPPA, 64 600 Anglet, France

**Keywords:** marine biomass, fishery by-products, green solvents, DES, solvents recyclability

## Abstract

*Polybius henslowii*, an abundant yet unexploited species of swimming crab, was investigated as a potential source of α-chitin and calcium lactate using deep eutectic solvents (DES) as extracting solvents. Choline chloride–malonic acid (CCMA) and choline chloride–lactic acid (CCLA) were used to obtain high purity α-chitin from ball-milled *P. henslowii* exoskeleton in 2 h at 120 °C, with yields of 12.05 ± 2.54% and 12.8 ± 1.54%, respectively. The physical and chemical characteristics of the obtained chitins were assessed using CHN elemental analysis, attenuated total reflectance–Fourier transform infrared spectroscopy, thermogravimetric analysis, and scanning electron microscopy. Furthermore, the CCLA solvent was reusable three times with little effect on the extract purity, and calcium lactate was produced at the end of the recycling cycles. The ensuing calcium lactate was also characterized in terms of chemical and physical properties. The obtained chitin is a promising raw material for downstream processing and the double valorization pathway with the obtention of calcium salts may increase the viability of a DES-based approach for the processing of mineralized substrates.

## 1. Introduction

Chitin is an abundant natural polymer, a polysaccharide composed of linear chains of poly(β-(1-4)-*N*-acetyl-D-glucosamine) presenting ordered crystalline microfibrils. It is emerging as a multifunctional product for the preparation of diverse advanced materials: papers [1], adsorbents for remediation of dye-contaminated wastes [2,3,4], reinforcing agent for composite biomaterials [5,6,7,8,9], biomedical devices [10,11,12], and biomimetic materials [13]. As a material, it has numerous advantageous qualities: biocompatibility and safety for human health [14,15], biodegradability [16,17], mechanical resistance [10,18,19], and suitability for chemical and physical modifications [20,21,22]. In particular, α-chitin is the predominant form of chitin found in the ocean, as a major constituent of the crustacean cuticle, among many others [23]. In this way, it can be derived from fishery by-products or side-streams [24], which are more economical than other sources, such as fungal production [25], and its production does not compete with food.

*Polybius henslowii* (Leach, 1820) is a highly abundant species of portunid crabs occurring off the Atlantic coasts of Europe and North Africa. Discards can reach up to 1240 tons/year [26], although portunid crabs have high survival rates when rejected [27]. Due to the obstruction of fishing gear and the damage they incur on it, they are largely regarded as a nuisance to fishing activity by professionals in the sector. Despite occurring on continental shelves within reach of small-scale coastal fisheries, it currently is not valorized at a large scale, only as bait [28] or fertilizer [29]. Recently, Avelelas et al. [30] have extracted chitin from this species via a conventional extraction approach and assessed the antifungal and antioxidant properties of the ensuing products. The chitin occurs in the species calcareous exoskeleton, associated with a mineral and protein-rich biomatrix [31] that must be removed to obtain the purified polymer.

The longest-standing methods of preparation of α-chitin involve a multiple-step process of demineralization by strong acids and protein removal through strong alkaline solutions that may generate high amounts of spent solvents and waste [32,33]. As such, the process is under active investigation for improved methods using biological approaches [34] or green chemistry approaches. One such approach is the use of deep eutectic solvents (DES) [35]. DES are alternative solvents that can be synthesized from renewable agro-resources. Eutectic signifies a lowered melting point, and DES are mixtures of two or more components that when combined, have a much-lowered melting point. They have many advantages beyond renewability, including reusability, biodegradability, nontoxicity, large-scale availability, low vapor pressure, thermal stability, low flammability, and are easy to prepare [35,36,37,38]. The eutectic nature of DES is due to the formation of extensive hydrogen bond networks between the components [39]. DES are predominantly mixtures of two components, one acting as a hydrogen bond acceptor, and the other as the donor. Choline chloride, which is an organic quaternary ammonium compound with a choline cation and chloride anion, is the most commonly used hydrogen bond acceptor in these systems [40,41,42]. Organic acids are also common components of DES [43,44], acting as hydrogen bond donors (HBD). Here, the focus will be directed on lactic acid and malonic acid. Lactic acid is a product of bacterial and mitochondrial metabolism and is produced in great quantity worldwide (270,000 tons/year) [45]. Malonic acid is another common organic acid produced by plants and is found in numerous fruits [46]. Both have been shown to be compatible with high-purity extraction of α-chitin from crustacean biomass, namely shrimps, prawns, and lobster shells, but never from crabs [47,48,49]. 

Furthermore, calcium salts of lactic acid (calcium lactate) also have good commercial potential as products of DES-mediated α-chitin extraction. Calcium lactate is a widely used additive in foods, authorized and commercialized under the E code E327, according to Regulation (EC) No 1333/2008. In the agricultural sector, it is also authorized for use in cattle, pig, and poultry feed [50]. It is also sold as a nutritional supplement in the US [51] and the EU (mixed with calcium gluconate and calcium carbonate) [52] and may be beneficial in the prevention of osteoporosis [53]. It can also be used as an additive to building materials to improve resistance [54]. Recently, other marine-derived raw materials, such as green crab exoskeletons [55] and mussel shells, have been explored as potential raw materials for the production of calcium lactate [56] by conventional extraction.

In this work, the objective was to pave the way for the use of *Polybius henslowii* as a raw material for the production of α-chitin and calcium salts with a DES-based, low-waste approach.

## 2. Results and Discussion

### 2.1. Optimization of the α-Chitin Extraction and Its Characterization

#### 2.1.1. Yields and Elemental Analysis of the Chitin-Based Extracts

The extraction yield, expressed as the weight of the obtained extracts over the raw material (see Equation (1) in Section 3.3), was used to evaluate the α-chitin extraction ability of the DES; the results were compared to α-chitin obtained with the conventional two-step extraction process (control). In the first step of the control process, the powdered crab shell was subjected to an acid treatment to remove minerals. In the second, alkali was used to remove proteins. This revealed a high proportion of acid-soluble material, predominantly mineral phase, in the shell, corresponding to 83.4 ± 1.1% and a relatively lower remaining portion of alkali-soluble material, representing only 4.7 ± 0.6% of the exoskeleton. The α-chitin yield was, thus, 12.9 ± 0.5% in the raw shell. Alvelelas et al. [30] obtained yields in the same range, circa 11.8%. 

The yields of the chitin-based extracts using DES as extracting solvents were superior to the yields obtained by the conventional method for the lower temperatures of 50 and 80 °C and for both of the DES and reaction times. Figure 1 displays yields of 16.7 and 19.9% at 50 °C and 14.9 and 18.5% at 80 °C that were indeed obtained for CCMA and CCLA, respectively, after a 1 h extraction. 

Furthermore, the results showed that one hour of treatment resulted in systematically higher yields (Figure 1), which indicated an insufficient reaction time for the complete removal of the undesirable compounds. This was further confirmed by the elemental analysis as the samples with high yields (i.e., low extraction temperatures and short extraction times) showed the highest relative amounts of nitrogen (between 7.5 and 8.5% for the temperatures of 50 °C and 80 °C for both CCMA and CCLA extracts (Table 1, which is coherent with a protein excess, compared to the traditional extraction (6.4 ± 0.2%).

The best results were obtained for the highest temperatures, 100 °C and 120 °C, for a 2 h extraction time with yields of 12.8 and 13.3% at 100 °C and 12.0% and 12.8% at 120 °C for CCMA and CCLA, respectively. Furthermore, the samples CCLA120-2 and CCMA120-2 showed the lowest %N of all chitin DES-extracts, with 6.9 ± 0.4% and 6.8 ± 0.1%, for CCMA and CCLA, respectively, although still slightly superior to that of the conventional method of 6.4 ± 0.2% (Table 1). These %N were concordant with the contents of a pure chitin that should contain between 6.9% (100% acetylated) and 6.0% (50% acetylated and close to chitosan) [57]. The high residual %N could be explained by the ineffective removal of proteins due to the inability of the acidic DES to disrupt protein–chitin linkages [58]. These results were consistent with previous studies that showed that higher temperatures, 100 °C and above, coupled with longer extraction times led to higher purity of chitin extracts when using DES [47,59,60]. The differences in yield can be explained by the viscosity of the DES, which may have been unable to penetrate deep into the chitinous matrix [60] at lower temperatures. 

Theoretically, carbon/nitrogen ratio (C/N) varies from 5.15 in chitosan to 6.86 in chitin, the fully N-acetylated biopolymer [61]. The C/N ratios obtained by elemental analysis listed in Table 2 showed that the two samples were very close to that of the theoretical pure chitin were CCMA120-2 and CCLA120-2 with C/N ratios of 6.41 and 6.76, respectively. The degree of acetylation (DA) of these two samples were found to be 73.1% and 94.1% for CCMA120-2 and CCLA120-2, respectively. The DA of the chitin extracted by using the conventional approach was 84.8%. Based on these results, CCMA120-2 and CCLA120-2 were selected for further characterization, namely the chemical structure, morphology, and thermal properties.

#### 2.1.2. Structural and Morphological Characterization of the Obtained α-Chitin Samples

The ATR-FTIR analysis revealed characteristic α-chitin spectra for all of the samples (Figure 2). The spectrum of the original crab shell powder is available in the Appendix A. Indeed all the samples showed two distinct sub-bands around 1652 cm^−1^ and 1620 cm^−1^ [62], corresponding to the amide I band vibrations, with this doubling being a notable typical feature of α-chitin. In addition to this characteristic, other features attributed to chitin were observable in the extracts, such as the amide II band and the amide III bands at 1550 cm^−1^ and 1307 cm^−1^, respectively. Additionally, –NH and –OH stretching between 3500 and 3100 cm^−1^ were visible, and a band at 2875 cm^−1^ corresponded to the C-H stretching vibrations.

Although the spectra obtained with the DES showed a strong similarity to the spectrum obtained with the conventional extraction method, it was noted that the spectra corresponding to the α-chitin samples extracted with DES showed a slight band around 1745 cm^−1^, corresponding to C=O stretching vibrations. This band could be related to the formation of strong interactions (e.g., H bonds) between the DES and target molecules, making them difficult to remove completely [35]. 

The WAXS spectra collected on the three samples (CCLA, CCMA, and conventional) displayed crystal reflections, corresponding to the α-chitin, according to the Minke and Blackwell model [63]. The major crystal reflections were 002, 012, 110, 101, 022, 121, 102, 013, 061 (Figure 3) that were also observed by Pacheco et al. [64]. The C.I. was determined from the peak, corresponding to the 110 reflection located at *q =* 1.35 Å^−1^ (Equation (3)). The C.I. were found to be 71.6% and 81.9% for CCMA and CCLA, respectively, which is in agreement with the DA results (i.e., CCLA presented higher DA). The C.I. of the conventional chitin was 80.6%, very close to CCLA values, and those by Férnández-Martin et al. [65] (around 80.3%) and Salaberria et al. (around 79%) [66].

The morphology of the samples’ powders was examined using SEM and is displayed in Figure 4. The samples were grinded by hand and the grain size was disparate. All samples showed the same flake-like shape that is common to α-chitin powder. The extraction techniques had no influence on the size or the morphological aspects of the powders.

#### 2.1.3. Thermogravimetric Analysis

The thermogravimetric analysis (TGA) was carried out to assess the thermal stability and degradation profiles of the different chitin extracts. All samples were tested both in nitrogen and air atmosphere. Whereas the former offers an inert environment to focus only on the main organic degradation pattern of chitin, the latter shows the complete degradation of organic matter, which means that the residue content can be interpreted as the total inorganic matter content [67]. These data, alongside ATR-FTIR results, can be correlated with the purity of the chitin extracted, which is key information for potential applications. The TGA and their derivative (DTG) profiles are shown in Figure 5, and Table 2 summarizes the most important thermal parameters for each sample. 

In a nitrogen atmosphere, the α-chitin samples first exhibited a slight loss between 30 °C and 120 °C. This was due to the evaporation of the residual water content of the samples. Then, around 200 °C, the main loss occurred with a single step degradation pattern, with a sharp peak around 380 °C. During this phase, different degradation reactions of chitin happen, such as polysaccharide depolymerization, thermal cracking, ring opening, and the deterioration of acetylated chitin units [68]. After 400 °C, the profile showed a very gentle slope up to 750 °C that can be referred to as the carbonization stage, where some of the previously formed biochar evaporates [69]. The remaining residue is then composed of one or two of the following, component inorganic matter (i.e., biominerals) and carbonized chitin, also referred to as char or biochar [67]. Although all three α-chitin extracts curves displayed similar patterns, slight variations could be observed. The initial degradation temperature (Td_i_) of conventional, CCMA120-2, and CCLA120-2 extracts were estimated around 229, 218, and 204 °C, respectively. Their maximal degradation temperatures (Td_max_) were 388, 377, and 371 °C, respectively. This showed a slight shift of the degradation peaks towards lower temperatures when using DES for chitin extraction. In comparison, Td_max_ for α-chitin has previously been reported in the range of 350–390 °C, depending on its biological origin, from 390 °C in grasshoppers [70] to 372 °C in crab shell [71], and 350 °C in shrimp shells [65]. Regarding residual mass at 750 °C, they were relatively similar with 15.02%, 16.94%, and 15.52% for the conventional, CCMA120-2, and CCLA120-2, respectively.

In an air atmosphere, the α-chitin samples underwent the same loss between 30 to 120 °C, corresponding to water evaporation. In this atmosphere, the main degradation was divided in two steps. First, as with nitrogen, the main part of chitin was degraded, resulting in a maximal degradation peak at around 300–330 °C. This degradation corresponded to the disintegration of polysaccharide chains and acetylated chitin units. Then, around 350 °C, a second phase began, with a maximum peak around 500 °C. This step represented the combustion reaction of the carbonized chitin, and thus the devolatilization of char. In the end, at 750 °C, almost no residue was left for all three samples. This, in addition with FTIR and elemental analysis, ensured that pure chitin was extracted without inorganic impurities. It also indicated that the 15–17% residual matter obtained in a nitrogen atmosphere was purely composed of residual char. Although the same shift regarding initial and maximal degradation temperatures was noticeable with the air atmosphere curves, there was no significant variation regarding the second degradation peak (Td_2_). This analysis showed that DES approaches for α-chitin extraction had little to no effect on the thermal properties and thermal degradation behaviors of the extracted chitin. 

### 2.2. Recycling of the DES

One of the major advantages of using the DES is that they can be re-used several times. Initial trials showed rather low recyclability for CCMA (data not shown). This may be due to the fact that malonic acid decomposes at high temperatures to produce acetic acid and CO_2_ [72]. Furthermore, the evaporation point of acetic acid is close to that of water in the conditions used for evaporation (40 mBar, 60 °C bath, 10 °C cooling), and some may have been lost in the recycling process. Therefore, from the above extraction processes CCMA120-2 and CCLA120-2, CCLA was selected for extensive recycling.

The CCLA was usable at 120 °C up to four times (initial extraction plus three recycles). Recycling was discontinued after the third round, as the solvent was excessively viscous and its recuperation became very difficult. The DES had also taken a golden-brown color due to protein and pigment build-up (the ATR-FTIR spectra of the DES after extraction and precipitation of the remaining chitin showed the appearance of a large band circa 1590 cm^−1^, [73] Appendix A). This result was coherent with the findings of Bisht et al. [47] who were also able to re-use the same solvent three times using crayfish shells at similar ratios (1:20 *w/w*) and similar temperatures (115 °C). Other authors have reported re-use up to five times using shrimp shells, higher ratios, and other DES but incurring significant drops in extraction efficacy and the loss of extract purity [74]. 

As displayed in Figure 6A, no significant variations in yield were observed from R0 to R3, with values from 12.8% for R0 to 14.2% for R3. Nonetheless, elementary analysis revealed that the %N increased after the first cycle, from 6.87% for R0 to 7.14% for R3 (Table 3). This increase could probably be related with a loss of the solvent capacity to remove proteins after R2. It was also observed that the DAs of R2 and R3 were lower than the two first cycles (Table 3).

The ATR-FITR spectra of the CCLA120-2 extracts obtained after all the recycling cycles (Figure 6B) showed characteristic α-chitin bands, as previously described in Figure 4.

### 2.3. Precipitation and Characterization of the Calcium Lactate

After the four cycles of chitin extraction using the CCLA DES at 120 °C and 2 h (CCLA120-2), calcium lactate was precipitated. In this part of the work, we studied the possibility of precipitating calcium lactate as a co-product of the chitin extraction because, as previously mentioned, calcium lactate is a widely used salt with a number of industrial applications.

The elemental analysis of the crystals obtained after the precipitation (Table 4) showed that the purity of the calcium lactate was 95.5% (based on the carbon content of the sample (31.53 ± 0.58) and reported to the theoretical content (33.02%) [56]). The presence of nitrogen in the precipitated sample suggested the possible presence of residual proteins or calcium nitrate. Nonetheless, this amount was very low (%N = 0.30 ± 0.02).

The aspect of the crystals was roughly spherical with a diameter approaching 1 mm in the larger crystals, and it was coralloid when adjoined (Figure 7). 

The chemical structure of the calcium lactate was confirmed by ATR-FTIR spectrum (Figure 7). The spectra showed high similarity with commercial calcium lactate and calcium lactate derived from mussel shells by Mititelu et al. [56]. In the spectra displayed in Figure 7, the large band among 3500 and 3157 cm^−1^ was attributed to the -OH stretching vibrations, the band at 2982 cm^−1^ was assigned to the C-H stretching vibrations, the two intense peaks at 1560 cm^−1^ and 1480 cm^−1^ were assigned to the asymmetric stretch band of the carboxylate group (large dipole moment), and the strong band observed at 1120 cm^−1^ was assigned to the C-O bond stretching [56].

The TGA thermogram showed several different steps of decomposition, similarly to what was observed in the work of Mititelu et al. [56]. The first step between 30 and 200 °C with a weight loss of 5.5% corresponded to the evaporation of associated H_2_O molecules (Figure 8). Following this initial weight loss, a second weight loss of 19% was observed at a Td_max_ of 291 °C, and another 30% between 370 and 510 °C. These two steps corresponded with the formation of mineral calcium carbonate (CaCO_3_). An additional step was observed at Td_max_ 540 °C, corresponding with a 6.9% weight loss, which could be attributed to the decomposition of anhydrous calcium nitrate (from residual choline chloride or proteins) at 500–600 °C, producing CaO and NO_2_ [75]. In the final step, between 613 °C and 805 °C, the CaCO_3_ was transformed to calcium oxide (CaO) via calcination [56,76]. The final residue was 28.9%, corresponding to the calcium oxide. 

In previous studies, ChCl-based DESs have been used to extract both hydrophilic and lipophilic compounds from marine biomass [77], such as macroalgae [78] or microalgae [79], for example. The exoskeleton of *Polybius henslowii* has been shown to contain proteins and lipids, in addition to chitin [30], which could also be co-extracted by the DESs and reduce the purity of the calcium lactate fraction. The N% of the calcium lactate extract was measured using elemental analysis. It showed that the extract contained around 0.30% N (data not shown). This result, coupled with the slight band observed around 1745 cm^−1^ on the ATR-FTIR, corresponding to C=O stretching vibrations, indicated that the calcium lactate extract presented some impurities. This calcium lactate could be used as an additive to bioconcrete, as inclusion of the material enhances self-healing of this type of material [54], even in aquatic environments [80]; however, further work to increase the purity of the fraction is necessary to extend application to fields, such as calcium supplements [56] or feed [50].

## 3. Materials and Methods

### 3.1. Chemicals and Raw Material

Malonic acid (propanedioic acid, C_3_H_4_O_4_, Alfa Aesar), lactic acid (2- hydroxypropanoic acid, C_3_H_6_O_3_, 90%, Fisher Chemical), and choline chloride (2-hydroxyethyl(trimethyl)azanium;chloride, C_5_H_14_ClNO, ACROS Organics) were purchased from Thermo Fisher Chemicals (France). All chemicals were reagent grade. 

Whole crabs (around 50 specimens) from the species *Polybius henslowii* were obtained from a local fishing boat, caught in the south of the Bay of Biscay in pot traps in 2020 and 2021. Crabs were collected and identified by C. McReynolds. Crabs were frozen at -25 °C before further processing. To obtain the crab exoskeletons (carapaces), the crabs were thawed at room temperature and the exoskeleton was removed. The exoskeletons were then washed under warm running tap water to remove surface impurities before being dried for at least 72 h at 60 °C. The dried exoskeletons were then ball-milled in a rotary ball-mill RETSCH PM100 (Haan, Germany) for 20 min at 400 rpm, with changes in direction every 4 min to ensure a homogenous powder. The powder was sieved through a 0.2 mm mesh and stored at 4 °C until use.

### 3.2. Preparation of the Deep Eutectic Solvents

Herein, the efficacy of two acidic DES systems were investigated on the extraction of α-chitin, as listed in Table 5. Precisely weighed dry choline chloride, a type of hydrogen bond acceptor (HBA), was mixed with malonic acid or lactic acid, both types of hydrogen bond donors (HBD), at a 1:2 molar ratio (HBA:HBD). For the preparation of the DES systems, the flasks were first set in a heating bath at 80 °C under magnetic agitation until viscous, transparent solutions were formed. The flasks were then transferred to a rotary evaporator (BUCHI R-100; Flawil, Switzerland), and excess water was evaporated overnight at 40 mBar and 60 °C.

### 3.3. Chitin Extraction

The extraction procedure of α-chitin was adapted from previous studies [48,60]. The powdered exoskeleton was introduced into 100 mL screw-cap vials with the DES in a 1:25 (*w*/*w*) ratio; the mixture was heated to 50, 80, 100, or 120 °C in a glycerol bath for 1 or 2 h (Table 1) under magnetic stirring (400 rpm). To quench the reaction, twice the solution volume of 80 °C water was added to the vial, and the mixture centrifuged at room temperature for 10 min at 4000 rpm to separate the precipitated chitin from the supernatant. The chitin was then rinsed a further 5–10 times with hot water (80 °C) until solution pH approached neutrality.

To compare the yields and properties of the chitin extracted using the DES, a conventional two-step demineralization and protein removal approach was used, with the method described by Avelelas et al. [30], who also extracted chitin and chitosan from *P. henslowii*, with slight modifications. Briefly, the powdered crab shell was immersed in a 1 M HCl solution in a 1:30 (*w*/*v*) ratio for 30 min at room temperature. It was then washed with distilled water and dried in an incubator for 24 h at 60 °C. The samples were then placed in a 1 M NaOH solution at a 1:30 (*w*/*v*) ratio, at 70 °C for 2 h in a glycerol bath, then washed until neutrality, and dried as stated previously.

The yields were determined gravimetrically following Equation (1):(1)$$Yield\;\left(\%\right)=\frac{Extract}{Dry\;carapace}\times 100$$

### 3.4. Recycling of the DES and Calcium Lactate Precipitation

The recycling scheme displayed in Figure 9 was undertaken in a multiple-step process. First, the excess water from the quenching solution was evaporated in a rotary evaporator (BUCHI, Switzerland) at 50 °C and 30 mPa. Ice-cold absolute ethanol was then added to the dry DES in a 5:1 (*w*/*w*) ratio in order to remove eventual contaminant remaining in the DES, such as proteins. The solution was centrifuged at 4000 rpm for 10 min at room temperature to collect precipitates, and then the EtOH was evaporated. The as-reconstituted DES was then re-used, as described above. After the third cycle, the reconstituted DES was highly viscous and further recycling was stopped. 

### 3.5. Calcium Lactate Precipitation

To precipitate calcium lactate, the extraction solution of the DES obtained after the final extraction was quenched with 1:1 (*w*/*w*) distilled water. This solution was placed at 4 °C for 48 h to allow for the precipitation of calcium lactate crystals. The crystals were filtered using a 0.2 mm mesh sieve and rinsed with ice-cold EtOH until colored surface contamination was removed. This precipitate was then dried at 60 °C for 24 h in a circulating air oven before further analysis. The ensuing precipitate was characterized in terms of structural composition using ATR-FTIR and elemental analysis and thermal stability by TGA analysis.

### 3.6. Characterization of the Samples

#### 3.6.1. CHN Elemental Analysis

To determine carbon (C), nitrogen (N), and hydrogen (H) contents of the chitin-based extracts and calcium lactate, a FLASHEA 1112 Elemental Analyzer (Delft, Netherlands) was used and the measurements were performed at least in duplicate. BBOT (2, 5-bis (5-tert-butyl-benzoxazol-2-yl)-thiopen, C_26_H_26_N_2_O_2_S, Thermo Fisher, Waltham, MA, USA) and sulfanilamide (4-aminobenzenesulfonamide, C_6_H_8_N_2_O_2_S, Thermo Fisher, Waltham, MA, USA) were used to calibrate the equipment using standards in linear calibration mode. An amount of 5 mg of vanadium pentoxide was mixed with 1.5 mg of each sample and incinerated at 900 °C on a He carrier gas. Eager 300 software was used to obtain CHN chromatograms and relative percentages of each atom. 

The degree of acetylation of the ensuing chitin was determined from the carbon/nitrogen ratio (*C*/*N*). This ratio varies from 5.145 in completely N-acetylated chitosan to 6.861, the fully N-acetylated biopolymer of chitin (C_8_H_13_O_5_N repeat unit). The degree of acetylation (*DA*) was calculated, according to Equation (2) [81]:(2)$$DA=\frac{\frac{C}{N}-5.145}{6.861-5.145}\times 100$$

#### 3.6.2. Attenuated Total Reflectance—Fourier Transform Infrared (ATR-FTIR) Spectrophotometry

The ATR-FTIR spectra of the chitin extracts and calcium lactate were acquired using a ThermoScientific Samrt iX Nicolet iS20 infrared spectrometer (ThermoScientific, Waltham, MA, USA) equipped with a KRS-5 crystal (refractive index 2.4; incidence angle 45°). The spectra were obtained from 500 to 4000 cm^−1^ in transmittance mode with a resolution of 4 cm^−1^ and after 128 scan accumulations.

#### 3.6.3. Scanning Electron Microscopy (SEM) 

Before scanning electron microscopy observation, chitin extracts were coated with Au at 30 mA for 60 s under vacuum using a DESK V Denton Vacuum, Moorestown, NJ, USA. Scanning electron micrographs were then taken using a HIROX SH-3000 (Hirox Europe, Limonest, France) operating at 25 Kv accelerated voltage.

#### 3.6.4. Wide Angle X-ray Scattering (WAXS)

The WAXS data were collected on the Swing beamline at SOLEIL (Saclay, France). The X-ray wavelength used was 1.033 Å, with a sample-to-detector distance of 0.1248 m. In the present paper, the scattering curves are plotted as a function of the scattering vector norm *q*, defined as *q* = 4*π*/*λ.*sin(*θ*) with 2*θ* the scattering angle. Standard data corrections were applied, i.e., background subtraction and sample thickness normalisation. Before collecting the data, the powdered samples were slightly pressed to obtain 7 mm pellets in order to be self-supported.

The crystallinity index (C.I.) was determined from Equation (3), according to the method of Focher et al. [82].
(3)$$C.I.\;\left(\%\right)=\frac{I_{110}-{\;I}_{\mathrm{am}}}{I_{110}}\times 100$$
where I_110_ is the peak corresponding to 110 reflection (arbitrary units) and I_am_ represents the intensity scattered by the amorphous phase at *q* = 1.135 Å ^−1^ (i.e., 2*θ* = 16°) [82]. 

#### 3.6.5. Thermogravimetric Analysis (TGA)

Thermogravimetric investigations were carried out using a thermal analysis system TGA 2, Mettler Toledo (USA). For chitin, a heating rate of 10 °C min^−1^ from 30 to 750 °C in both nitrogen and air atmosphere was used. For calcium lactate, a heating rate of 10 °C min^−1^ from 30 to 900 °C in air atmosphere was applied. Sample weights were 8–10 mg, and the gas flow rate was 60 mL min^−1^. The samples were stored at room temperature before testing and were not dried in an oven beforehand, thus, retaining their residual humidity.

## 4. Conclusions

Here, two DES, CCMA and CCLA, were used to determine their potential as green solvents for the extraction of α-chitin and calcium lactate from *P. henslowii* shells, opening the way for the potential valorization of an abundant and noncommercial crab species. Both solvents yielded the highest purity chitin, based on N% content, at temperatures of 120 °C for 2 h. The CCLA solvent was usable four times with a slight increase in chitin impurities. This may be seen as an additional opportunity, using a multiple-product approach as part of a bio-refinery concept. The possibility of precipitating calcium lactate by-products rendered the whole process more economically attractive, as the price of the raw materials for the DES is higher than that of conventional acid/alkali approaches. Moreover, the choline-rich supernatant could also be incorporated into feed—for example in aquaculture or in poultry feed, which would deserve further study. This work showed that using DES with a multiple-product approach could increase the value of fishery discards.

## Figures and Tables

**Figure 1 marinedrugs-20-00717-f001:**
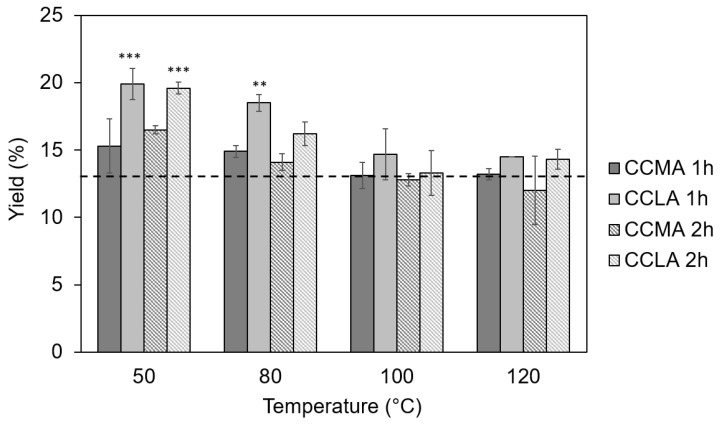
Yields of the α-chitin-based extracts from *P. henslowii*. The dotted line represents the yield obtained through the conventional approach (ALC). Stars indicate significant differences between the trial condition yields and that of the conventional approach (*** *p* > 0.001, ** *p* > 0.01, ANOVA with pairwise Turkey *t*-tests).

**Figure 2 marinedrugs-20-00717-f002:**
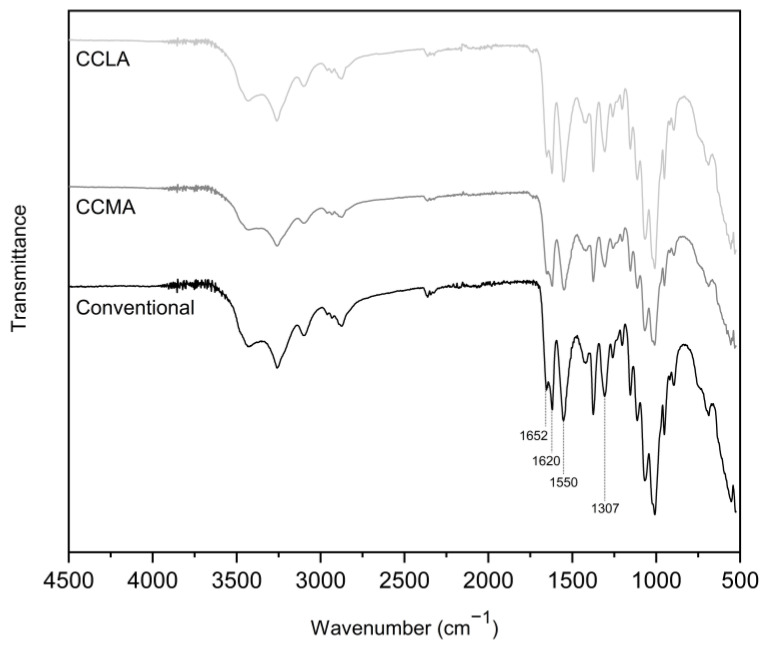
ATR-FTIR spectra of the α-chitin samples obtained with the previously selected DES and conventional extractions.

**Figure 3 marinedrugs-20-00717-f003:**
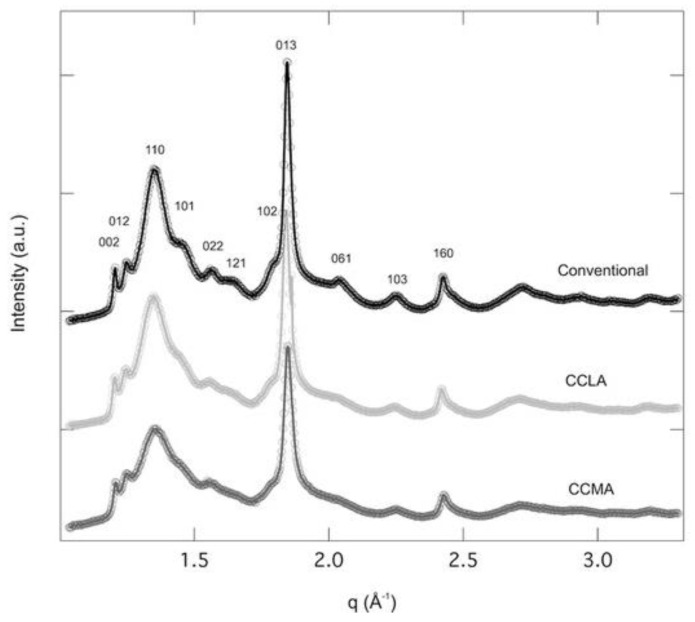
WAXS diffractograms of α-chitin samples extracted using conventional approach and DES, i.e., CCLA and CCMA.

**Figure 4 marinedrugs-20-00717-f004:**
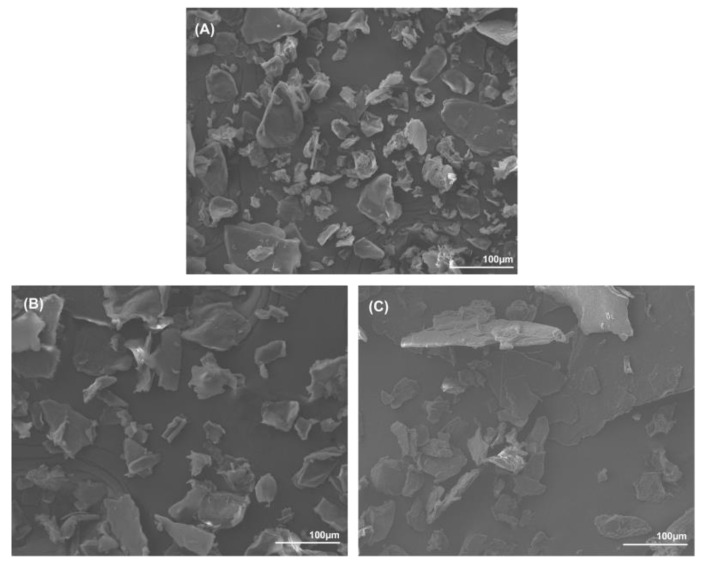
SEM pictures (×200) of chitin extracts by: (**A**) conventional approach; (**B**) CCLA120-2; and (**C**) CCMA 120-2.

**Figure 5 marinedrugs-20-00717-f005:**
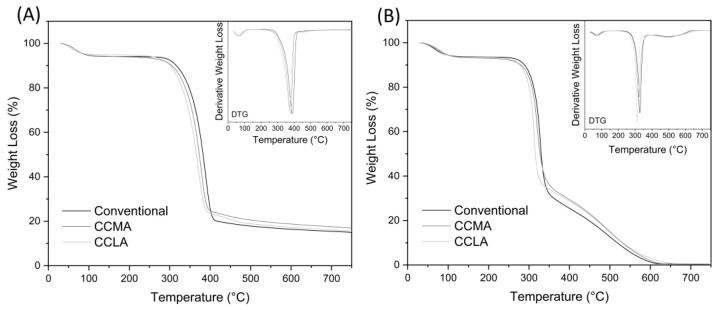
TGA and DTG profiles of the α-chitin extracts from conventional chitin, CCLA120-2 and CCMA120-2 chitin sample, in (**A**) nitrogen atmosphere and (**B**) air atmosphere.

**Figure 6 marinedrugs-20-00717-f006:**
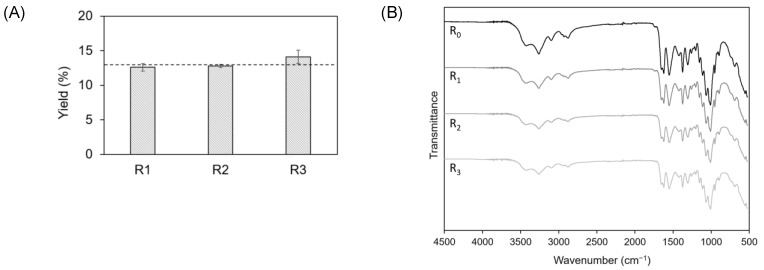
(**A**) Yields (dotted line represents CCLA120-2 yield (R0)); and (**B**) ATR-FTIR spectra of the CCLA120-2-based extracts in the different recycling cycles.

**Figure 7 marinedrugs-20-00717-f007:**
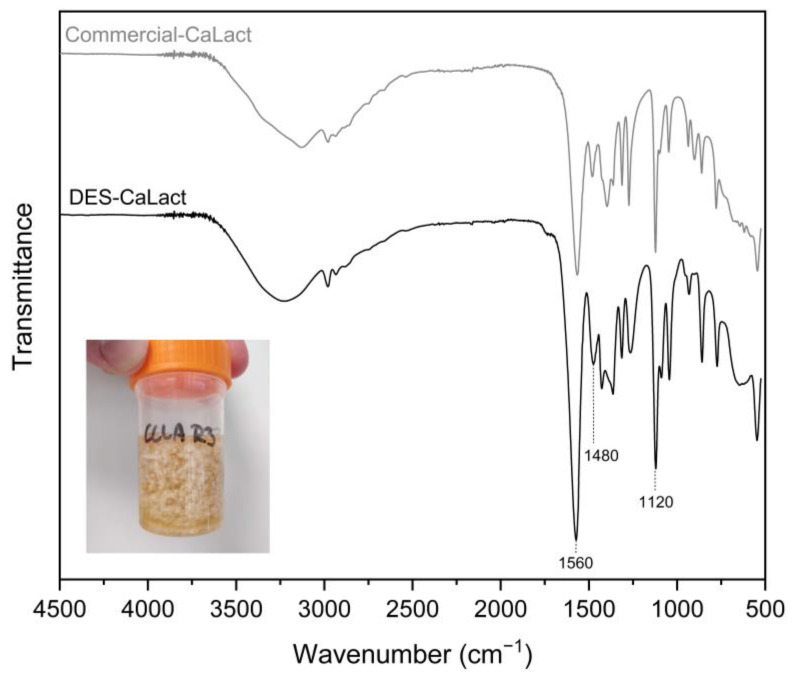
ATR-FTIR spectra of calcium lactate (CaLact) obtained after DES extraction at R3 (general aspect is displayed), in comparison with commercial calcium lactate.

**Figure 8 marinedrugs-20-00717-f008:**
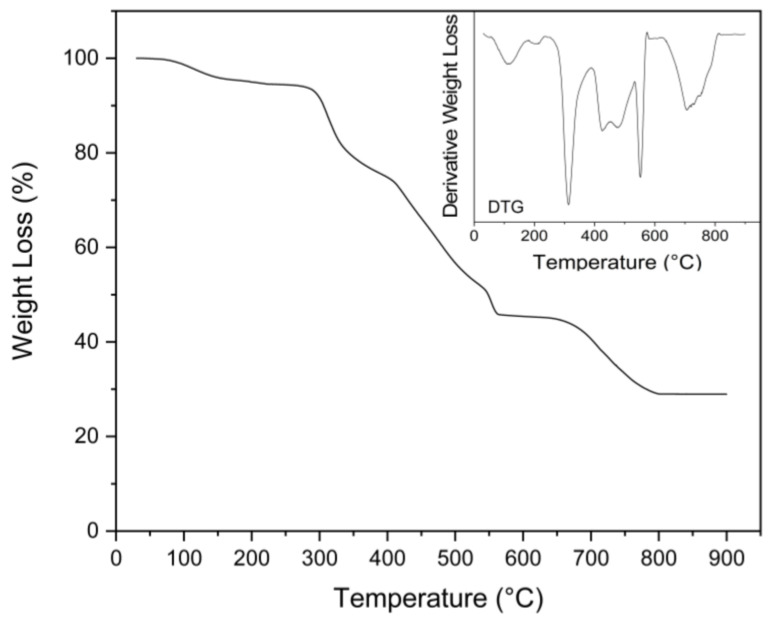
TGA and DTG profiles of obtained calcium lactate.

**Figure 9 marinedrugs-20-00717-f009:**
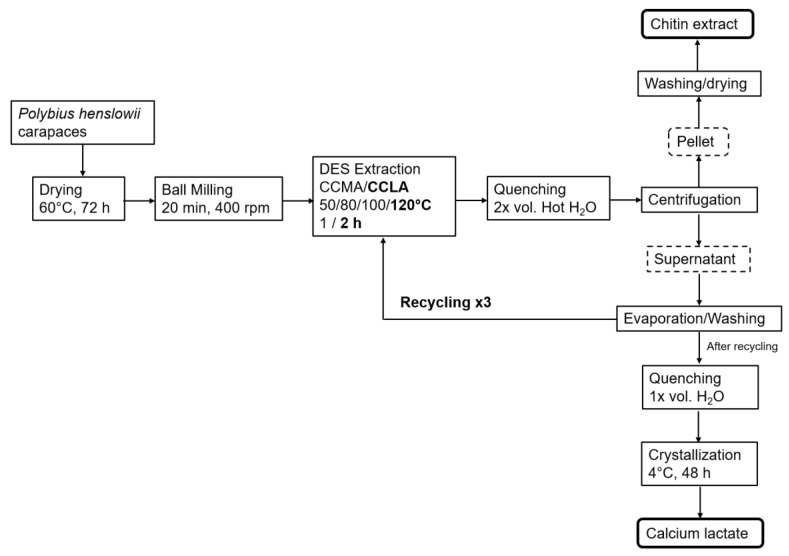
Schematic approach of the chitin extraction and DES recycling.

**Table 1 marinedrugs-20-00717-t001:** CN elemental composition of the chitin-based extracts.

Sample Identification	T (°C)	C/N	N (%)
Conventional	NA	6.60	6.43 ± 0.15
CCMA 1 h	50	5.63	7.89 ± 0.05
	80	5.68	7.86 ± 0.15
	100	6.11	7.38 ± 0.25
	120	6.33	7.05 ± 0.15
CCMA 2 h	50	5.68	7.86 ± 0.15
	80	5.77	7.59 ± 0.42
	100	5.89	7.21 ± 0.52
	120	6.41	6.83 ± 0.07
CCLA 1 h	50	5.36	8.27 ± 0.03
	80	5.17	8.53 ± 0.07
	100	5.87	7.61 ± 0.00
	120	6.20	7.17 ± 0.45
CCLA 2 h	50	5.24	8.56 ± 0.79
	80	5.75	7.80 ± 0.42
	100	6.03	7.46 ± 0.02
	120	6.76	6.87 ± 0.35

**Table 2 marinedrugs-20-00717-t002:** Initial temperature of degradation (Td_i_), maximum degradation temperatures (Td_max_), secondary degradation temperatures (Td_2_), and residual content of the obtained chitin at 750 °C.

	N Atmosphere			Air Atmosphere			
Chitin Sample	Td_i_ (°C)	Td_max_ (°C)	Residual (%)	Td_i_ (°C)	Td_max_ (°C)	Td_2_ (°C)	Residual (%)
Conventional	229	388	15.02	216	331	497	0.49
CCLA120-2	204	371	15.52	201	308	493	0.54
CCMA120-2	218	377	16.94	208	313	494	0.32

**Table 3 marinedrugs-20-00717-t003:** Elementary analysis, C/N ratio, and DA of CCLA120-2 chitin extracts for the different recycling cycles.

Recycling Cycles	C/N	N (%)	H (%)	DA (%)
R0	6.76	6.87 ± 0.31	6.43 ± 0.18	94.1
R1	6.72	6.75 ± 0.08	6.56 ± 0.08	91.8
R2	6.42	7.04 ± 0.09	6.59 ± 0.08	74.3
R3	6.37	7.14 ± 0.23	6.57 ± 0.06	71.1

**Table 4 marinedrugs-20-00717-t004:** Measured and theoretical elemental composition of calcium lactate.

Composition	C (%)	H (%)	N (%)	Ca^2+^ (%)
Measured in this work	31.53 ± 0.58	4.83 ± 0.04	0.30 ± 0.02	NM
Theoretical values of calcium lactate	33.02	4.62	0.00	18.37

**Table 5 marinedrugs-20-00717-t005:** Description of the DES systems (i.e., type, molar ratio, and abbreviation), and α-chitin extraction conditions: temperatures and sample identification (1 and 2 correspond to the extraction time in hours).

HBA	HBD	Molar Ratio (HBA:HBD)	DES Abbreviation	Reaction Temperature (°C)	Sample Identification
Choline chloride	Malonic acid	1:2	CCMA	50	CCMA50-1 CCMA50-2
				80	CCMA80-1 CCMA80-2
				100	CCMA100-1 CCMA100-2
				120	CCMA120-1 CCMA120-2
Choline chloride	Lactic acid	1:2	CCLA	50	CCLA50-1 CCLA50-2
				80	CCLA80-1 CCLA80-2
				100	CCLA100-1 CCLA100-2
				120	CCLA120-1 CCLA120-2

## Data Availability

Not applicable.

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
