# Peer review of "Double Valorization for a Discard—α-Chitin and Calcium Lactate Production from the Crab Polybius henslowii Using a Deep Eutectic Solvent Approach"

_marinedrugs, 2022, doi:10.3390/md20110717_

Round 1

Reviewer 1 Report

The manuscript entitled “Double valorization for a discard - α-chitin and calcium lactate production from the crab Polybius henslowii using a deep eutectic solvent approach" presents the development of a DES-based extraction method for the separation of α-chitin and posterior precipitation of calcium lactate from the remaining extract. Different DES and extraction temperatures are studied, considering the extraction yield and nitrogen content of the extract as the key variables determining the quality of the separation of α-chitin, and posterior obtention of calcium lactate crystals by precipitation.  The work provides interesting insights on the extraction process, as well as different characterization steps to ensure the identity of the target compound, although some deficits can be improved before considering it for publication, as mentioned in the following comments:

·         Introduction. An interesting addition would be information on the composition of the exoskeleton besides chitin and calcium lactate, for the reader to have an idea on what kind of compounds are eliminated by the use of the applied DES (or by the acid and alkaline steps of the conventional method).

·         Section 2. Minor details, such as the use or not of spaces in the expression of errors (1±0.1 or 1 ± 0.1?), should be revised.

·         Section 2.1.1. The yield definition should be given here, instead than on section 3.3.

·         Table 1. How were the errors for the C/N ratio calculated? For example: How is it possible to get a 0.00 error for C/N from %C and %N measurements with 0.19% and 0.15% errors?

·         Table 1. Is the %C data necessary in this table, considering that analysis is performed only based on the %N and C/N ratio?

·         Figure 5. The DTG inserts should have larger numbers and letters, to improve visualization.

·         Section 2.1.3/Introduction. Why is the thermal analysis of the obtained chitin relevant? Perhaps some information of the introduction would be a good addition to the manuscript, since it would highlight the importance of the results obtained in Section 2.1.3.

·         Section 2.2. Although not critical considering the aim of this work, some characterization of the DES (i.e., FTIR spectra) as the recycling cycles advance would be a valuable addition in this section.

·         Section 2.2 – Line 243. What do the changes in DA mean regarding the use of a recycled DES? Does this mean that a different fraction of chitin is extracted? Or is it an error stemming from the fact that more protein seems to be extracted (as suggested by the %N data)? If the former is true, aren’t the DA results somewhat tainted?

·         Figure 7 (A). The picture and structure (is the latter strictly necessary?) should be bigger to improve their visualization.

·         Table 5. The authors should consider placing this table in a position before the first mention of the different DES and extraction conditions, given that the sample identifications defined in the table are used throughout the manuscript (and are never explicitly defined therein).

Author Response

Reviewer 1

The manuscript entitled “Double valorization for a discard - α-chitin and calcium lactate production from the crab Polybius henslowii using a deep eutectic solvent approach" presents the development of a DES-based extraction method for the separation of α-chitin and posterior precipitation of calcium lactate from the remaining extract. Different DES and extraction temperatures are studied, considering the extraction yield and nitrogen content of the extract as the key variables determining the quality of the separation of α-chitin, and posterior obtention of calcium lactate crystals by precipitation.  The work provides interesting insights on the extraction process, as well as different characterization steps to ensure the identity of the target compound, although some deficits can be improved before considering it for publication, as mentioned in the following comments:

Introduction. An interesting addition would be information on the composition of the exoskeleton besides chitin and calcium lactate, for the reader to have an idea on what kind of compounds are eliminated by the use of the applied DES (or by the acid and alkaline steps of the conventional method).

Thank you for your suggestion. We added a sentence on the subject on line 49, p2.

Section 2. Minor details, such as the use or not of spaces in the expression of errors (1±0.1 or 1 ± 0.1?), should be revised.

Thank you for the attention to detail, a space has been added where missing.

Section 2.1.1. The yield definition should be given here, instead than on section 3.3.

Thank you for insightful comment, the opening sentence was modified as follows (line 89 p2): The extraction yield, expressed as the weight of the obtained extracts over the raw material (see Equation 1. Section 3.3. below) was used to […]

Table 1. How were the errors for the C/N ratio calculated? For example: How is it possible to get a 0.00 error for C/N from %C and %N measurements with 0.19% and 0.15% errors?

This was reported erroneously, the ratio should not include the errors. This has been rectified and the errors removed. Many thanks.

Table 1. Is the %C data necessary in this table, considering that analysis is performed only based on the %N and C/N ratio? 

As per your suggestion, the %C data has been removed for better clarity and conciseness.

Figure 5. The DTG inserts should have larger numbers and letters, to improve visualization. 

Thank you for your comment, the DTG inserts have been modified accordingly.

Section 2.1.3/Introduction. Why is the thermal analysis of the obtained chitin relevant? Perhaps some information of the introduction would be a good addition to the manuscript, since it would highlight the importance of the results obtained in Section 2.1.3.

Thank you for your comment. We report this information in the beginning of the discussion section 2.1.3, and as per your suggestion we have added some information to accentuate the importance of this data (line 186-187 p8).

Section 2.2. Although not critical considering the aim of this work, some characterization of the DES (i.e., FTIR spectra) as the recycling cycles advance would be a valuable addition in this section.

Thank you for your suggestion. We have included the FTIR spectra of the DES in a Supplementary Information file (Figure S2) mentioned in the text (Line 241 p 9).

Section 2.2 – Line 243. What do the changes in DA mean regarding the use of a recycled DES? Does this mean that a different fraction of chitin is extracted? Or is it an error stemming from the fact that more protein seems to be extracted (as suggested by the %N data)? If the former is true, aren’t the DA results somewhat tainted?

Thank you for your comment. Indeed, the increase in N seems to indicate that there is more protein, which would throw off the DA as this is calculated according to the N content. As such we have decided that reporting this value is not coherent and have removed it.

Figure 7 (A). The picture and structure (is the latter strictly necessary?) should be bigger to improve their visualization.

Thank you for your suggestion, Figure 7 has been modified to be more explicit.

Table 5. The authors should consider placing this table in a position before the first mention of the different DES and extraction conditions, given that the sample identifications defined in the table are used throughout the manuscript (and are never explicitly defined therein).

Thank you for your suggestion. We agree with your comment, nonetheless, we have respected the journal format. We believe that the readers can consult the Materials and Methods section in order to verify the samples identification.

Reviewer 2 Report

I have read the manuscript and I have a few questions and recommendations.
1. In section 3.1, you must specify who identified Polybius henslowii, as well as the number of the voucher and the place where it was stored.
2. For the data in Figure 1, it is necessary to make a statistical comparison with the conventional method. Based on this figure, all data are not statistically different from each other.
3. For section 3.3, please specify what was calculated according to formula 1: chitin, chitosan, or total extract from the exoskeleton of Polybius henslowii?
4. In section 3.5, the formula for calculating the yield of calcium lactate is not correct. It is necessary to make a calculation taking into account the chemical conversion of calcium phosphate from the exoskeleton of Polybius henslowii to calcium lactate/malate, respectively.
5. In section 2.3, it is necessary to present the FITR spectra of the reference calcium lactate and compare it with that obtained from the exoskeleton.
6. "The yield calculated based on the conversion of crab shell to calcium lactate..." must be reported taking into account the chemical conversion.
7. It is known from recent publications that NADES extract both lipophilic and hydrophilic substances (eg https://doi.org/10.3390/molecules26144198 etc.). What precipitates with calcium lactate during extraction. What was washed with ice-cold ethanol? How much ethanol was used for flushing?
8. In Section 2.3, the purity of your calcium lactate should be determined using a standard method. This will allow us to draw conclusions about the prospects of the your approach.

Author Response

Review 2

I have read the manuscript and I have a few questions and recommendations.

  1. In section 3.1, you must specify who identifiedPolybius henslowii, as well as the number of the voucher and the place where it was stored.

Thank you for your suggestion. These informations were added in the revised manuscript.

  1. For the data in Figure 1, it is necessary to make a statistical comparison with the conventional method. Based on this figure, all data are not statistically different from each other.

Thank you for your suggestion. We added the statistical comparison with the conventional method on figure 1.

  1. For section 3.3, please specify what was calculated according to formula 1: chitin, chitosan, or total extract from the exoskeleton ofPolybius henslowii?

Thank you for the insight, this has been rectified in equation 1. We prefer to use the term total extract before verifying the chemical characteristics of the chitin.

  1. In section 3.5, the formula for calculating the yield of calcium lactate is not correct. It is necessary to make a calculation taking into account the chemical conversion of calcium phosphate from the exoskeleton ofPolybius henslowiito calcium lactate/malate, respectively.

Thank you for your comment. We were not able to do this modification. To do it properly, we have removed the sentence about the yield in section 2.3. and in the equation used in section 3.5. The aim of this work was to show that it was possible to valorize a second product from crab exoskeleton, and we have demonstrated it by ATR-FTIR, EA and TGA. 

  1. In section 2.3, it is necessary to present the FITR spectra of the reference calcium lactate and compare it with that obtained from the exoskeleton.

As per your suggestion, we have included the ATR-FTIR spectra of commercial calcium lactate in Figure 7 along with the spectra obtained from our extracts.

  1. "The yield calculated based on the conversion of crab shell to calcium lactate..." must be reported taking into account the chemical conversion.

Please see the answer giving in point 4.

  1. It is known from recent publications that NADES extract both lipophilic and hydrophilic substances (eg https://doi.org/10.3390/molecules26144198etc.). What precipitates with calcium lactate during extraction. What was washed with ice-cold ethanol? How much ethanol was used for flushing?

Thank you for your comment. We agree that the calcium lactate could contain impurities such as proteins or lipids; however, these were not analyzed or quantified. The CHN and TGA analysis suggests the presence of some contaminants in the precipitated calcium lactate. To address this issue, a paragraph was added in section 2.3. (see question 8).

As explained in 3.4. the DES post-precipitation and drying was washed with ice-cold ethanol, at 5:1 w/w ratio in an attempt to remove contaminants.

  1. In Section 2.3,the purity of your calcium lactate should be determined using a standard method.This will allow us to draw conclusions about the prospects of the your approach.

Thank you. We agree with you that in order to increase the possible applications of our calcium lactate, a standard method should be used to determine the purity of the extract.

We therefore added a paragraph to address these issues (points 7 and 8) in section 2.3.

“In previous studies, ChCl-based DESs have been used to extract both hydrophilic and lipophilic compounds (doi.org/10.1016/j.gee.2019.01.012) from marine biomass, such as macroalgae (https://doi.org/10.3390/molecules26144198) or microalgae (https://doi.org/10.1016/j.biortech.2016.05.120) for example. The exoskeleton of Polybius henslowii has been shown to contain, in addition to chitin, proteins and lipids (https://doi.org/10.3390/md17040239) which could also be co-extracted by the DESs and reduce the purity of the calcium lactate fraction. The N% of the calcium lactate extract was measured using elemental analysis. It showed that the extract contained around 0.30% N (data not shown). This result, coupled with the slight band observed around 1745 cm-1 on the ATR-FTIR corresponding to C=O stretching vibrations, indicated that the calcium lactate extract presented some impurities.This calcium lactate could be used as an additive to bioconcrete, as inclusion of the material enhances self-healing of this type of material (https://doi.org/10.3390/su13169269), even in aquatic environments (https://doi.org/10.1016/j.conbuildmat.2021.122332), however further work to increase the purity of the fraction is necessary to extend application to fields such as calcium supplements (10.3390/md20010045)  or feed (https://doi.org/10.2903/j.efsa.2017.4938)”.

Round 2

Reviewer 2 Report

The authors have made the necessary corrections and I have no more questions.